# PixPerfect: Seamless Latent Diffusion Local Editing with Discriminative Pixel-Space Refinement

Haitian Zheng[1]*  Yuan Yao[2]*  Yongsheng Yu[2]
Yuqian Zhou[1]  Jiebo Luo[2]  Zhe Lin[1]

[1]Adobe Research  [2]University of Rochester

{hazheng, yuqzhou, zlin}@adobe.com,
yynobug@gmail.com, yyu90@ur.rochester.edu, jluo@cs.rochester.edu

## Abstract

Latent Diffusion Models (LDMs) have markedly advanced the quality of image inpainting and local editing. However, the inherent latent compression often introduces pixel-level inconsistencies, such as chromatic shifts, texture mismatches, and visible seams along editing boundaries. Existing remedies, including background-conditioned latent decoding and pixel-space harmonization, usually fail to fully eliminate these artifacts in practice and do not generalize well across different latent representations or tasks. We introduce PixPerfect, a pixel-level refinement framework that delivers seamless, high-fidelity local edits across diverse LDM architectures and tasks. PixPerfect leverages (i) a differentiable discriminative pixel space that amplifies and suppresses subtle color and texture discrepancies, (ii) a comprehensive artifact simulation pipeline that exposes the refiner to realistic local editing artifacts during training, and (iii) a direct pixel-space refinement scheme that ensures broad applicability across diverse latent representations and tasks. Extensive experiments on inpainting, object removal, and insertion benchmarks demonstrate that PixPerfect substantially enhances perceptual fidelity and downstream editing performance, establishing a new standard for robust and high-fidelity localized image editing.

## 1 Introduction

Image inpainting [28, 13, 3] and local editing [66, 42, 55] aim to modify a specified image region according to high-level instructions, such as a text prompt [57] or a reference image [6, 41] while ensuring the coherence with surrounding pixels. As an atomic operation for interactive image editing, it plays a fundamental role for applications ranging from object removal, insertion, to creative content regeneration. Recent advances in text-to-image generation models, particularly latent diffusion models (LDMs) [38], have driven remarkable progress in image inpainting [13, 3, 20] and local editing [7, 41, 55]. These methods perform diffusion processes [18] in a low-dimensional and semantically compressed latent space [12] and demonstrate impressive capacity in generating complex and semantically coherent visual content guided by textual or structural cues.

However, latent local editing methods often struggle to maintain pixel-level consistency between synthesized regions and their surrounding context [47]. Specifically, the latent encoding and decoding inherent in these approaches often introduces low-level compression errors, such as color and textures mismatch, hindering the pixel-level matching with the original background. Furthermore, when the edited region is pasted-back into the source image, small visual differences usually become more pronounced, producing visible mismatches at the boundaries. Such artifacts are usually hard to eliminate. In fact, our experiment found that more expressive latent representations, such as the 16-channel VAE in FLUX [3], often exacerbate local editing artifacts due to poor generalization

during diffusion inference. Such visual artifacts, as shown in Fig. 4 and Fig. 5, are a mixture of: i) chromatic shifts at editing boundaries, ii) misaligned textures, inconsistent noise or grain patterns, and iii) visible seams arising from content discontinuities and they typically persist across diverse inpainting and local editing methods, posing a fundamental challenge that undermines both perceptual fidelity and practical usability across a wide spectrum of editing applications.

Several approaches have been explored to mitigate local editing artifacts. First, latent-space modifications integrate background information during generation: Asymmetric-VQGAN enriches the latent decoder with partial background inputs for context-aware decoding [65], and ASUKA introduces color augmentation during decoder training to simulate uniform chromatic discrepancies [47]. Second, post-hoc pixel-level harmonization methods refine the synthesized region after generation: naive Poisson blending solves for seamless gradient transitions [36], while DiffHarmony++ employs a learning-based harmonization model for pixel-level adjustment [63]. Despite these advancements, subtle hue shifts, texture mismatches, and content discontinuities often persist for those methods, as human perception remains sensitive to minute visual discrepancies at editing boundaries. Moreover, the reliance on a specific latent space [65, 47] limits the generalization of these methods to alternative representations and diverse editing scenarios.

In this work, we identified three fundamental challenges associated with local editing artifacts and proposed **PixPerfect** as a general-purpose, high-fidelity, and pixel-level artifact removal framework for inpainting and local editing. Our proposed framework addresses three unsolved issues:

(i) **Subtle visual difference:** how to eliminate persistent and cumbersome boundaries artifacts caused by subtle but perceptible pixel-value difference or noise-pattern discrepancies?

(ii) **Complexity of local-editing artifacts:** how to handle the complexity of the artifacts that mixes chromatic shifts, inconsistent noise or grain patterns, and content discontinuities?

(iii) **Generalization:** how to develop a unified solution that generalizes across diverse latent diffusion models, latent spaces [38, 3] or applications?

Consequently, we propose three novel contributions to address those issues. First, we propose a novel **discriminative pixel space** that transforms the RGB color space into a more discriminative representation where subtle hue and texture mismatches between an edited region and its background become more perceptible. Such a differentiable transformation is incorporated as a loss to enhance the model sensitivity for more precise color and texture alignment. Second, we design a **comprehensive data pipeline** that simulates local editing artifacts in a realistic setting. Our data pipeline simulates a diverse set of inpainting degradations including non-uniform color shifts, texture inconsistencies, noise pattern variations, and content discontinuities, thereby enabling the refiner to learn across multiple failure modes. Third, we formulate the refinement process as **pixel-level refinement** as opposed to latent decoding compared to prior works [65, 47], yielding a general-purpose solution for inpainting and local-editing artifact removal.

With extensive evaluations and visualizations across a diverse set of inpainting and local editing models, we demonstrate that the refiner can robustly correct local editing artifacts and consistently improve the performance and visual quality of a wide range of existing methods. As such, our approach yields higher quantitative scores and visibly superior results, and noticeably, it significantly boosts a wide spectrum of state-of-the-art methods for image inpainting, object removal and object insertion. All these evidence shows the effectiveness and necessity of our proposed scheme for achieving high-quality image inpainting and localized editing.

## 2  Related Work

**Image Inpainting and Local Editing.**   Image inpainting and local editing are confined to masked regions. GAN-based approaches [34, 54, 60, 4, 43, 28, 13] achieve structure recovery by learning semantic priors, often enhanced by attention [53, 31, 37] and multi-scale feature fusion [56]. Diffusion-based methods, including RePaint [32], Stable Diffusion inpainting [38], and FLUX-Fill [3], demonstrate strong inpainting capabilities but may produce inconsistencies around mask boundaries or hallucinate unfaithful content. Beyond pure inpainting, local editing extends the scope to broader semantic manipulations. Works like BrushNet [20] and CLIPAway [11] adopt masked diffusion pipelines guided by CLIP or user input, while OmniPaint [55] and FreeCompose [7] support both object removal and insertion. Other methods explore region-specific editing via retrieval-based

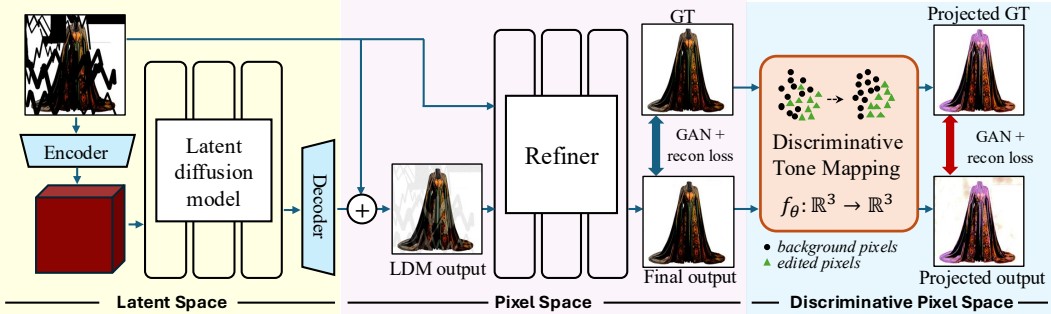

Figure 1: Given a partially composited image generated by latent diffusion model, PixPerfect refines the output to correct harmonization artifacts between the synthesized region and the background. A discriminative color-space transformation is employed to enhance sensitivity to subtle chromatic and textural discrepancies, thereby improving color harmonization and texture coherence.

augmentation [6, 42, 41], layout control [29], or hybrid latent/image-space optimization [1, 2]. Despite promising results, latent-diffusion based methods still struggle to seamlessly blend the edits and maintain local consistency. We propose the universal method PixPerfect to solve this issue.

**Improving Image Details for Diffusion Models.** Latent diffusion models [38, 3] compress images with VAEs [25], causing loss of fine details. To solve this issue, various works have explored enhancing latent representations. Consistency-aware training [40] and frequency-aware VAEs [49] aim to preserve structure during compression, while decoder-based improvements [12, 33, 65] target the reconstruction phase. In small domains, post-generation methods [5, 46] provides enhancement for generated contents. Masked generative priors [4, 47, 48] offer better fidelity by supervising denoising with partial reconstructions. However, these approaches still fall short of full consistency [48, 47]. We argue ensuring consistency in pixel space is a more refined approach and present a pixel space refiner.

**Image Harmonization and Refinement.** Image harmonization aims to correct appearance inconsistencies between a composited foreground and its background. Early learning-based methods [44, 8, 30] cast harmonization as an image-to-image translation problem, adjusting the foreground's illumination and color to match the context. Attention-based and multi-scale designs [9, 15, 24] improve blending along object boundaries. Recent works introduce contrastive learning [16], transformer-based harmonization [19], and diffusion models [64, 27] semantic shifts. In image enhancement, generic correctors such as ARCNN for JPEG artifacts [10], lens aberration correction [52], and deep sharpening networks [45, 26] further improve realism. However, existing harmonization methods are typically designed for manually composited inputs where the foreground is a complete, well-defined object pasted into a clean background. In contrast, diffusion-based inpainting involves arbitrary-shaped masked regions with potentially complex semantics. Moreover, many harmonization models focus on aligning the foreground appearance, but do not explicitly enforce background consistency, making them less suitable for correcting seams introduced by latent diffusion. Our work bridges this gap by addressing both semantic restoration and foreground-background consistency within a unified refinement framework tailored to latent diffusion artifacts.

## 3 Methods

Inpainting and local editing aim to modify a region of an original image $\boldsymbol{x}_{\mathrm{ori}}$, yielding an edited result $\boldsymbol{x}_{\mathrm{gen}}$ that alters only pixels within a binary mask $\boldsymbol{m} \in {0, 1}^{H \times W}$ while preserving background content. LDM-based approaches frequently introduce inconsistencies along mask boundaries following latent decoding and background compositing. To overcome this limitation, PixPerfect employs a pixel-level refinement network implemented as a generative adversarial network (GAN) [14], denoted as G, to restore pixel-level coherence. The refiner produces

$$\boldsymbol{x}_{\mathrm{pred}} = \mathrm{G}(\boldsymbol{x}_{\mathrm{gen}}, \boldsymbol{m}),$$

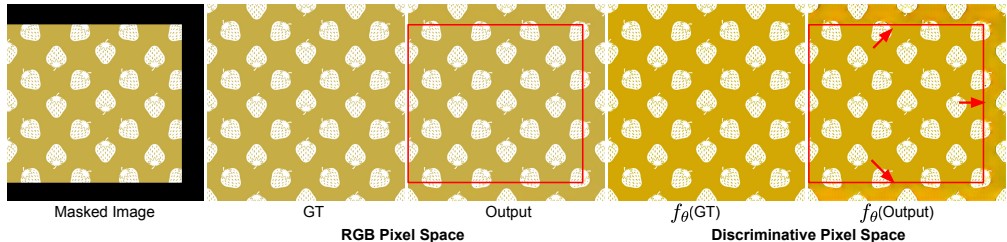

| Masked Image | GT | Output | $f_\theta$(GT) | $f_\theta$(Output) |
|:---:|:---:|:---:|:---:|:---:|
| | **RGB Pixel Space** | | **Discriminative Pixel Space** | |

Figure 2: The effect of discriminative pixel space on enhancing subtle background mismatch.

such that $\boldsymbol{x}_{\mathrm{pred}}$ aligns closely with the pixel-consistent oracle image $\boldsymbol{x}_{\mathrm{gt}}$ in both the edited region and its surroundings.

## 3.1 The Discriminative Pixel Space

Precise alignment of color and texture between synthesized regions and the surrounding background is essential for achieving pixel consistent and seamless results. In the inpainting and image harmonization literature, this objective is commonly enforced via minimizing a combination of $\ell_1$ loss, perceptual losses and mask-conditioned adversarial loss on the pixel-space to constrain chromatic and structural similarity,

$$\mathcal{L}_{\mathrm{pixel-space}} = w_1 \cdot \|\boldsymbol{x}_{\mathrm{pred}} - \boldsymbol{x}_{\mathrm{gt}}\|_1 + w_2 \cdot \|\phi(\boldsymbol{x}_{\mathrm{pred}}) - \phi(\boldsymbol{x}_{\mathrm{gt}})\|_1 + w_3 \cdot \mathrm{D}(\boldsymbol{x}_{\mathrm{gen}}, \boldsymbol{m}), \quad (1)$$

where $\phi(\cdot)$ denotes perceptual feature extractor [58], D is the mask-conditioned adversarial loss and $w$ are balancing weights.

However, refinement networks trained with only these loss frequently produce outputs with subtle yet persistent hue shifts, mismatched texture or noise patterns, and visible seams that remain detectable by expert observers during high-fidelity editing. In fact, this issue arises across GAN-based inpainting [43, 60, 61], image harmonization [63], and latent decoding methods [65, 47] across various settings.

We argue that such visual distortions are due to the insensitivity of the conventional pixel-space objectives to subtle color or textures misalignments. To overcome this limitation, we introduce a discriminative pixel space that *amplifies perceptual discrepancies between the synthesized region and its background*. Specifically, we define a discriminative tone mapping function $f_\theta \colon \mathbb{R}^3 \to \mathbb{R}^3$ parameterized by $\theta$, which transforms pixel value vector $\boldsymbol{x}[p] \in [0,1]^3$ at each location $p$ into a discriminative color space, thereby rendering color and texture mismatches more salient.

The visual effect of the tone mapping $f_\theta$ is illustrated in Fig. 2. After applying $f_\theta$, both the prediction and the ground-truth are projected into the discriminative pixel space via $\boldsymbol{y}_{\mathrm{pred}} = f_\theta(\boldsymbol{x}_{\mathrm{pred}})$ and $\boldsymbol{y}_{\mathrm{gt}} = f_\theta(\boldsymbol{x}_{\mathrm{gt}})$, respectively, where subtle color and texture mismatches between foreground and background are amplified. Accordingly, we define the discriminative pixel space loss as the combination of $\ell_1$, perceptual and conditioned adversarial loss using the same weighting:

$$\mathcal{L}_{\mathrm{disc-space}} = w_1 \cdot |\boldsymbol{y}_{\mathrm{pred}} - \boldsymbol{y}_{\mathrm{gt}}\|_1 + w_2 \cdot \|\phi(\boldsymbol{y}_{\mathrm{pred}}) - \phi(\boldsymbol{y}_{\mathrm{gt}})\|_1 + w_3 \cdot \mathrm{D}'(\boldsymbol{y}_{\mathrm{gen}}, \boldsymbol{m}). \quad (2)$$

Designing the tone mapping function $f_\theta$ is critical to enable the discriminative pixel space. In principle, $f_\theta$ must be differentiable, computationally lightweight, and adaptive to individual samples at training stage. To satisfy these requirements, we parameterize $f_\theta$ as a polynomial regression, yielding a closed-form, sample-specific, differentiable mapping. Specifically, the regression is defined as:

$$\boldsymbol{y}_c = \sum_d p_{c,d} \boldsymbol{x}_c^d, \quad (3)$$

for each channel $c \in \{R, G, B\}$, $D$ is the polynomial degree and $\theta = (p_{c,1}, \cdots, p_{c,D})$ are the parameters. To facilitate content discrimination, the regression inputs are defined as the pixel values of the predicted image $\boldsymbol{x}_{\mathrm{pred}}$ and regression targets are specified by an image that amplifies the hue difference between $\boldsymbol{x}_{\mathrm{pred}}$ and $\boldsymbol{x}_{\mathrm{gt}}$ within the composition mask:

$$\boldsymbol{y}_{\mathrm{amp}} = \boldsymbol{x}_{\mathrm{gt}} + \beta\left(\boldsymbol{x}_{\mathrm{pred}} - \boldsymbol{x}_{\mathrm{gt}}\right),$$

where $\beta > 1$ controls the amplification strength. For the implementation, the Moore–Penrose pseudoinverse [35] is employed to compute the regression coefficients. To improve training stability,

**Non-uniform Color Shifting**

**Texture Pattern Mismatch**

**Content Discontinuity**

Color jittering    Gradient blending

$x_{\text{sim}} = \alpha \cdot x_{\text{img}} + (1 - \alpha) \cdot x_{\text{jit}}$

Apply different texture in the hole and background

JPEG

Jpeg artifact    Noise pattern

Smoothing    VAE distortion

Expand mask   Inpainter   Seamless on boundary   *Seam*

Composite the original background

Figure 3: **Artifact Simulation Pipelines.** Each module illustrates one of the three simulated artifact types used to train our refiner. **(Left)** Non-uniform color shifting is created via local jittering and gradient blending. **(Middle)** Texture-pattern mismatches are simulated through JPEG artifacts, noise, and VAE decoding distortions applied selectively inside or outside the mask. **(Right)** Content discontinuity is emulated by expanding the mask boundary and applying off-the-shelf inpainting followed by background recomposition.

we apply balanced sampling to select an equal number of pixels inside and outside the mask, and $\beta$ is drawn uniformly from $[20, 40]$. Finally, the outputs are clamped to valid ranges after tone mapping.

As such, our overall training objective is a combination of the original pixel-space loss and the discriminative pixel-space loss:

$$\mathcal{L} = \mathcal{L}_{\text{pixel-space}} + \mathcal{L}_{\text{disc-space}}. \tag{4}$$

### 3.2 Simulating Local Editing Artifacts

Training a high-quality refiner requires diverse, controllable supervision that reflects the degradations introduced by LDM-based inpainting and editing. Relying on real diffusion outputs for supervision poses two challenges: (i) artifact distributions vary across models and prompts, hindering the construction of a consistent training set; and (ii) the ground-truth image $x_{\text{gt}}$ for a partially edited output $x_{\text{gen}}$ is often unavailable or ambiguous, particularly when large semantic gaps or content hallucinations occur.

To overcome these limitations, we design a synthetic artifact simulation pipeline that generates training pairs $(x_{\text{gen}}, x_{\text{gt}})$ by injecting controlled degradations into clean images. Unlike prior work [47], our simulation reproduces the complex, realistic local editing artifacts observed in diffusion outputs. Specifically, the pipeline applies a mixture of non-uniform color shifts, texture-pattern inconsistencies, content-mismatch discontinuities, soft and hard boundary effects, and autoencoder reconstruction artifacts. Each degradation module operates exclusively within masked regions, preserving background integrity and providing a clear learning signal for harmonization. Please refer to the supplementary material for further details.

**Non-uniform Color Shifting.** Inpainting and local editing often introduce chromatic discrepancies relative to surrounding background, which can be especially pronounced over heterogeneous regions (e.g., skylines). To model these effects, a non-uniform color augmentation pipeline is devised as opposed to [47]. First, uniform color shifts are simulated by applying random color jitter within the masked region [22]. Next, non-uniform chromatic variations are synthesized by alpha-blending two independently color-jittered versions of the input using a randomly generated gradient mask. This gradient alpha map produces spatially varying hue and luminance shifts that closely mimic realistic color artifacts. The alpha-blending process can be formulated as

$$x_{\text{sim}} = \alpha \cdot x_{\text{img}} + (1 - \alpha) \cdot x_{\text{jit}}. \tag{5}$$

**Simulating Texture-Pattern Mismatch.** Diffusion-based inpainting often yields region-specific texture distortions—such as blurring, inconsistent noise patterns, smoothed details, the absence of background JPEG compression artifacts, and altered texture distributions due to latent decoding. To replicate these defects, the simulation pipeline introduces independent texture transformations within and outside the masked region. Specifically, the masked region undergoes random VAE reconstructions [38, 3] and Gaussian smoothing, while the background is subjected to JPEG compression artifacts. In addition, separate stochastic texture synthesis processes are applied to foreground and background, generating distinct noise and detail characteristics that mimic real-world texture mismatches.

**Simulating Content Discontinuities.** Partial compositing of generated content onto existing backgrounds can induce slight misalignments or content discrepancies at mask boundaries, yielding visible seams that compromise object integrity. To emulate this artifact, an off-the-shelf inpainting method [61, 60] reconstructs a narrow band straddling the mask edge, perturbing pixels on both sides of the boundary. The original background pixels are then composited back into the masked region, producing training examples that exhibit realistic boundary discontinuities similar to those introduced by diffusion-based edits. This simulation supplies the refiner with explicit examples of seam artifacts, enabling targeted correction.

**Mixing Soft and Hard Boundary.** Latent diffusion local edits often produce both feathered (soft) or abrupt (hard) seams that may be offset from the true composition boundary. To simulate these artifacts, we randomly perturb the composition mask for artifact generation with morphological dilation and erosion, then apply Gaussian blur with random kernel size to generate soft transitions. Blending content with these augmented masks enables our pipeline to generalize across both smooth and sharp seam artifacts.

### 3.3 Noise-adding and Inference-time Pooling

Following the recent GAN-based approach [21], we apply moderate Gaussian noise augmentation to the input pixels to stabilize the GAN training. Furthermore, as inference-time scaling has recently been recently proven to be effective for multiple domains related LLM and GenAI models. Inspired by this idea, we propose a simple yet effective inference-time pooling tricks to further boost the performance. Our intuition is by apply the refiner on different color jittering variation of the input image and perform pooling, a better refiner output can be produced. Specifically, given an initial image $x_{\text{gen}}$, we propose $N$ random color jittering inside the mask, resulting $x_{\text{gen}}^{(i)}$ for $i \in 1, \cdots, N$. Then we apply the refiner to the jittered image, and we propose to use the difference between input and refiner output $\|x_{\text{pred}}^{(i)} - \text{G}(x_{\text{gen}}^{(i)}, m)\|_1$ as an indicator to determine how well the input image is close to the grouth-truth. Finally, the best refiner output is selected as $x_{\text{pred}}^{(i^*)}$ whereas $i^* = \arg\min_i \|x_{\text{pred}}^{(i)} - \text{G}(x_{\text{gen}}^{(i)}, m)\|_1$.

## 4 Experiments

### 4.1 Experiment Settings

**Implementation details** Our model is built upon the CMGAN architecture [61] and trained on a curated dataset of approximately 300 million images at 1024×1024 resolution. Optimization uses Adam with a learning rate of 0.0005 and a batch size of 32. We interoperate larger perceptual and l1 weight, i.e. $w_1 = 64, w_2 = 5, w_3 = 1$ to enforce color consistency. the perceptual loss is computed using LPIPS [58] following [12]. For the tone mapping function, the maximal polynomial degree is set to $D = 5$ to avoid overfitting. Training is performed on a cluster of 32 NVIDIA A100 GPUs within one week. Further details are provided in the supplementary material.

**Evaluation Dataset** We evaluate PixPerfect on three major tasks—inpainting, object removal, and object insertion. For inpainting, we follow prior works and use two standard datasets: Places2 [62] and MISATO [47]. Places2 is a large-scale scene-centric dataset from which we randomly sample 2000 validation images and apply irregular masks of varying shapes and sizes to simulate occlusions. MISATO consists of 2000 512×512 images, each paired with a generated mask, specifically curated for evaluating semantic inpainting. For object removal, we use the RORDS dataset [39], which

| Dataset | MISATO | | | | | | Places2 | | | | | |
|---|---|---|---|---|---|---|---|---|---|---|---|---|
| Method | FID↓ | LPIPS↓ | L1↓ | PSNR↑ | U-IDS↑ | P-IDS↑ | FID↓ | LPIPS↓ | L1↓ | PSNR↑ | U-IDS↑ | P-IDS↑ |
| SDv1.5 [38] | 18.15 | 0.229 | 0.068 | 19.01 | 9.55 | 4.03 | 21.45 | 0.270 | 0.088 | 17.22 | 11.34 | 5.04 |
| **SDv1.5-PixPerfect** | 13.25 | 0.171 | 0.044 | 20.40 | 17.24 | **10.89** | 18.91 | 0.228 | 0.067 | 18.07 | 15.05 | 8.82 |
| SDv2 [38] | 18.68 | 0.236 | 0.067 | 19.04 | 8.24 | 3.83 | 21.13 | 0.271 | 0.086 | 17.25 | 11.16 | 5.29 |
| **SDv2-PixPerfect** | 16.28 | 0.189 | 0.048 | 19.81 | 13.13 | 7.71 | 19.12 | 0.231 | 0.069 | 17.90 | 15.45 | 8.82 |
| FLUX-Fill [3] | 14.66 | 0.195 | 0.062 | 20.90 | 8.39 | 3.18 | 19.05 | 0.240 | 0.074 | 19.33 | 7.89 | 3.12 |
| FLUX-Fill-AsyVQ [65] | 15.99 | 0.202 | 0.057 | 20.91 | 7.46 | 3.33 | 18.28 | 0.244 | 0.073 | 19.07 | 14.49 | 8.47 |
| FLUX-Fill-DH [63] | 14.02 | 0.190 | 0.056 | 20.89 | 10.38 | 4.79 | 18.18 | 0.236 | 0.071 | 18.99 | 10.48 | 4.74 |
| **FLUX-Fill-PixPerfect** | **10.87** | **0.141** | **0.036** | **22.18** | **18.09** | 9.53 | **15.61** | **0.194** | **0.052** | **20.04** | **19.08** | **11.69** |

Table 1: Quantitative comparison on MISATO and Places2. Our method substantially improves upon existing inpainting approaches and significantly reduces the FID score for FLUX-Fill [3].

contains 500 image pairs with human-annotated foreground masks and corresponding clean background ground-truths. For object insertion, we evaluate on a dataset of 300 triplets, each comprising a background image, a foreground object, and a ground-truth composite image.

**Benchmark** We evaluate the quality of generated images using a comprehensive set of metrics covering both perceptual similarity and pixel-level accuracy. These include FID[17] for distributional alignment, LPIPS[58] for perceptual similarity, L1 and PSNR for reconstruction fidelity, and P-IDS / U-IDS [60] to assess perceptual discriminability. In addition, for the object insertion task where ground truth composite image is not reliable, we report no-reference image quality scores such as MUSIQ [23] and MANIQA [51] to reflect global perceptual coherence. Together, these metrics provide a balanced evaluation across visual quality, semantic consistency, and low-level accuracy.

## 4.2 Comparison results

**Inpainting** PixPerfect is evaluated on latent diffusion-based inpainting models including SDv1.5, SDv2, and FLUX-Fill, across MISATO and Places2 datasets in Tab.1. PixPerfect consistently improves both perceptual (FID, LPIPS, IDS) and pixel-level (L1, PSNR) metrics over all baselines. Notably, the FLUX-Fill-PixPerfect achieves new state-of-the-art results. We further compare PixPerfect with the decoder-based method Asymmetric VQ-GAN[65] and the harmonization-based method DiffHarmony++ [63]. On both datasets, PixPerfect outperforms these methods by a clear margin across all metrics, demonstrating its effectiveness in correcting diffusion-induced artifacts.

Qualitative comparisons in Fig. 4 further highlight the differences. We paste back original unmasked regions to reveal editing artifacts. Both Asymmetric VQ-GAN and DiffHarmony++ show noticeable color shifts or texture inconsistency, particularly around mask boundaries or semantically complex regions. In contrast, PixPerfect produces sharper transitions and better-aligned textures, yielding more coherent and natural completions.

**Object removal** We evaluate PixPerfect on four representative diffusion-based object removal models: BrushNet [20], CLIPAway [11], PowerPaint [66], and OmniPaint [55]. Tab. 2 shows the results on the RORDS dataset. PixPerfect consistently improves all baselines across FID, LPIPS, L1, and PSNR, confirming its effectiveness across a diverse range of architectures and scenes. Even for the strongest model OmniPaint, PixPerfect significantly reduces the FID from 23.05 to 18.87 and improves PSNR from 24.67 to 27.96, demonstrating its capacity to refine already plausible outputs and further enhance structural fidelity and perceptual quality. Fig. 5 shows a representative example where editing artifacts overlap with soft shadows on reflective surfaces. PixPerfect eliminates the artifacts while preserving natural cues, resulting in coherent and visually consistent completions. These results highlight the robustness of PixPerfect in complex real-world removal scenarios and its utility as a general refinement module across diverse generative pipelines.

**Object Insertion** We evaluate PixPerfect on four representative diffusion-based object insertion models: Pbe [50], ObjectStitch [41], AnyDoor [6], and OmniPaint [55]. Tab. 3 reports quantitative results across both full-reference metrics (FID, LPIPS, L1) and no-reference perceptual scores (MUSIQ, MANIQA). PixPerfect consistently improves all baselines across all metrics, reflecting better structural alignment and visual fidelity after refinement. The gains are particularly large in models with more evident compositional artifacts, such as Pbe and ObjectStitch. Even in more

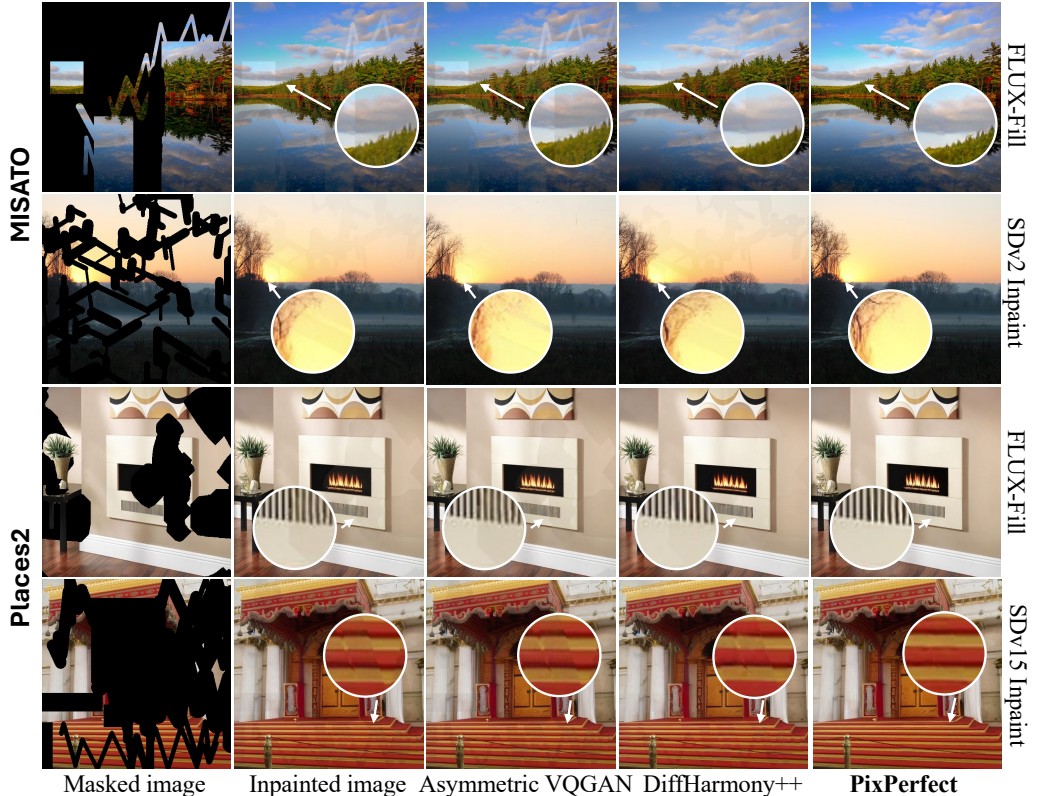

Figure 4: Qualitative comparison of inpainting results on MISATO [47] and Places2 [62]. PixPerfect consistently restores coherent structure and appearance across mask boundaries, and produces more accurate color and texture transitions compared to Asymmetric VQGAN [65] and DiffHarmony++ [63].

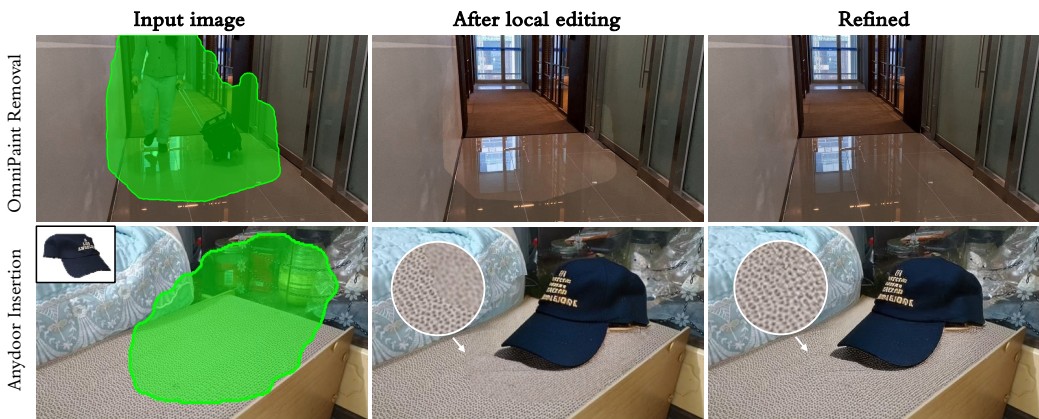

Figure 5: PixPerfect improves visual coherence of the local editing method OmniPaint [55]. In the removal case, it removes the hue artifact while restoring realistic reflections and shadows. In the insertion example, it aligns fine-grained textures around the boundary, producing seamless integration between the inserted object and its background.

| Method | FID↓ | LPIPS↓ | L1↓ | PSNR↑ |
|---|---|---|---|---|
| BrushNet [20] | 148.99 | 0.224 | 0.0604 | 20.00 |
| + PixPerfect | **144.54** | **0.152** | **0.0390** | **21.31** |
| CLIPAway [11] | 63.69 | 0.193 | 0.0593 | 20.78 |
| + PixPerfect | **54.78** | **0.113** | **0.0337** | **23.04** |
| PowerPaint [66] | 53.33 | 0.170 | 0.0613 | 20.75 |
| + PixPerfect | **43.40** | **0.097** | **0.0321** | **23.50** |
| OmniPaint [55] | 23.05 | 0.094 | 0.0420 | 24.67 |
| + PixPerfect | **18.87** | **0.060** | **0.0206** | **27.96** |

Table 2: Object removal results.

| Method | FID↓ | LPIPS↓ | L1↓ | MUSIQ↑ | MANIQA↑ |
|---|---|---|---|---|---|
| Pbe [50] | 97.53 | 0.269 | 0.0856 | 69.33 | 0.4746 |
| + PixPerfect | **91.21** | **0.236** | **0.0793** | **71.04** | **0.5070** |
| ObjectStitch [41] | 89.14 | 0.264 | 0.0838 | 69.27 | 0.4051 |
| + PixPerfect | **86.74** | **0.238** | **0.0790** | **71.38** | **0.5060** |
| AnyDoor [6] | 73.17 | 0.251 | 0.0794 | 68.53 | 0.4306 |
| + PixPerfect | **71.74** | **0.223** | **0.0764** | **71.53** | **0.5058** |
| OmniPaint [55] | **56.80** | 0.186 | 0.0713 | 70.32 | 0.5029 |
| + PixPerfect | 57.42 | **0.181** | **0.0678** | **71.54** | **0.5066** |

Table 3: Object insertion results.

advanced models, PixPerfect yields measurable improvements, indicating that subtle inconsistencies are still prevalent in modern diffusion-based insertion pipelines and can be effectively resolved through targeted refinement. Fig. 5 shows an insertion example the edited image suffers from severe texture mismatch. PixPerfect is capable of restoring textures that are consistent with the background.

### 4.3 Ablation Studies

To validate the effectiveness of each component in PixPerfect, we conduct an incremental ablation study based on FLUX-Fill, using the MISATO dataset for evaluation. Starting from the baseline, we progressively integrate each proposed module and report the results in Tab. 4. Applying a simple *paste-back* operation, which restores the unmasked regions from the original image, already yields substantial reductions in LPIPS and L1. This confirms that background distortion is a key source of degradation in latent diffusion-based inpainting.

Introducing the *refiner*—our base architecture for artifact suppression—further improves FID and LPIPS, though the gains remain modest without additional guidance. The key performance leap comes from adding the *enhancement loss* in the proposed discriminative pixel space. This transformation amplifies subtle chromatic and structural inconsistencies that are often overlooked in standard pixel space, enabling the refiner to align texture and color more precisely. As a result, all metrics improve significantly.

Ablation studies are further conducted on the variant of the discriminative pixel-space design. First, the influence of the polynomial degree in 3 is examined by varying the degree value during training. Unless otherwise stated, our default degree is set to $d = 6$. Empirical observations indicate that a low degree such as $d = 2$ leads to only limited tonal correction, reducing the loss effectiveness, whereas excessively high degrees such as $d = 10$ tend to overemphasize local details and generate outputs that deviate from the natural image distribution thus degrading performance. In addition, we examined a high-dimensional discriminative space variant that applies the discriminative transformation to VGG16 feature maps before computing the LPIPS loss. From the experiment, this variant achieves competitive performance, suggesting that our method is generalizable to a higher-dimensional discriminative space. However, its performance is slightly worse than the original pixel-space implementation, which we hypothesize is due to the loss of spatial precision along editing boundaries caused by spatial downsampling of feature maps. Furthermore, an alternative pixel-space enhancement loss is evaluated by employing HAAR decomposition to separate both the prediction and ground truth into low- and high-frequency components. Independent $\ell_1$ losses are then applied to each band, with a larger weight assigned to the low-frequency term to emphasize subtle chromatic variations. However, this design yields inferior results, as band-wise reweighting is difficult to integrate with perceptual or adversarial objectives, which are essential for high-quality image synthesis. Finally, incorporating the *inference-time pooling* strategy further stabilizes the predictions, mitigating noise and improving overall visual coherence.

## 5 Limitations and Broader Impacts

PixPerfect is a refinement module designed to correct low-level artifacts in diffusion-based inpainting and local editing. While effective in improving color consistency and texture alignment, it cannot correct major semantic errors from the generative model. Its performance depends on the availability of reasonably accurate initial predictions and pre-defined edited regions.

| Method | FID↓ | LPIPS↓ | L1↓ |
|---|---|---|---|
| FLUX-fill [3] | 14.6585 | 0.1950 | 0.0621 |
| + paste-back | 14.4022 | 0.1701 | 0.0395 |
| + refiner | 13.9874 | 0.1698 | 0.0402 |
| + enhance loss (d=6, default) | 10.9014 | 0.1425 | 0.0365 |
|    enhance loss (d=2) | 11.2244 | 0.1431 | 0.0362 |
|    enhance loss (d=10) | 11.0018 | 0.1407 | 0.0361 |
|    enhance loss on VGG features | 11.0525 | 0.1421 | 0.0360 |
|    Haar-based re-weighted loss | 11.3816 | 0.1431 | 0.0375 |
| + inference time pooling (**PixPerfect**) | **10.8675** | **0.1414** | **0.0363** |

Table 4: Ablation study on the MISATO dataset. Each component of PixPerfect progressively improves inpainting quality across perceptual and pixel-wise metrics.

PixPerfect can support a wide range of socially beneficial applications, such as accessible photo editing, digital restoration of historical or damaged media, and assistive tools for creators with limited visual or technical expertise, by improving visual fidelity and reducing editing artifacts. However, similar with other generative tools, it may also be misused for producing more convincing manipulated content. The method does not introduce new identity or demographic biases, but it inherits any biases present in upstream diffusion models and training datasets.

## 6 Conclusion

PixPerfect is presented as a general-purpose refinement module designed to rectify harmonization failures in LDM-based inpainting and local editing. By integrating a discriminative pixel-space objective, a realistic artifact simulation pipeline, and a direct pixel-level refinement framework, Pix-Perfect effectively mitigates chromatic shifts, texture misalignments, and boundary seams exhibited by a variety of diffusion-based editing models. Comprehensive evaluations across multiple tasks and architectures demonstrate substantial gains in perceptual fidelity and quantitative performance. Furthermore, as a lightweight, plug-and-play component, PixPerfect generalizes robustly to diverse editing scenarios. These findings highlight the potential to a pixel-consistent and reliable local image editing pipeline.

# Appendix

Supplementary Material for "PixPerfect: Seamless Latent Diffusion Local Editing with Discriminative Pixel-Space Refinement"

## A  The Latent Space Spatial Disentanglement Issue

Latent diffusion models operates on a compact latent space. However, the latent space are spatially entangled and not suitable for pixel-wise tasks. In this section, we study the latent space disentanglement issue.

Latent diffusion models encode images into a compressed latent space with an autoencoder. However, this latent representation lacks spatial disentanglement, limiting its suitability for fine-grained local editing. To illustrate this issue, we design a controlled experiment shown in Fig. 6. We encode both the original image and its masked counterpart using FLUX VAE [3], then construct a hybrid latent by combining the unmasked background from the masked input with the masked region from the original. This ensures that the latent representation differs only within a small localized area.

If the VAE decoder preserved spatial locality, such a localized change would not affect the reconstruction outside the masked region. However, the decoded image exhibits global shifts in background appearance, even where latent features remain unchanged. This behavior highlights a fundamental limitation of the latent space: local modifications can induce unintended global effects due to entangled representations. These observations motivate our refinement strategy, which operates in the pixel space to preserve spatial locality and ensure coherent integration between edited and unedited regions.

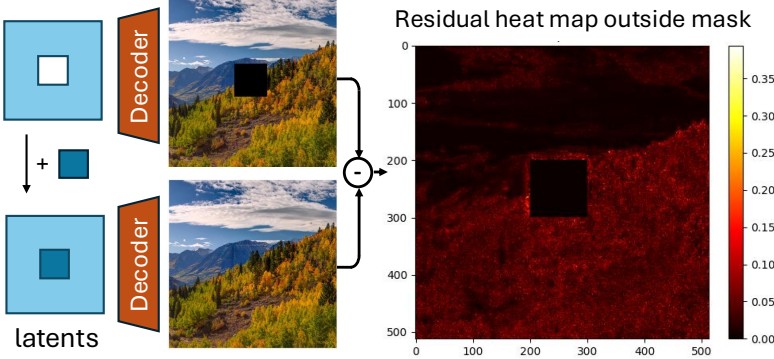

Figure 6: Replacing only the masked region in latent space leads to background drift in the decoded image, suggesting spatial entanglement in latent-based inpainting.

## B  More Experiments

### B.1  Efficiency Analysis

While improving visual consistency and perceptual fidelity is the primary goal of our refinement framework, it is also critical that the added refinement stage does not significantly increase the overall runtime. To this end, we analyze the computational cost of PixPerfect in comparison to the underlying latent diffusion sampling process.

Our refiner operates as a single-stage feed-forward network in the pixel space, and introduces negligible overhead compared to the iterative denoising procedure of diffusion models. For example, when applied to a 512×512 image on a single NVIDIA A100 GPU, the diffusion sampling with FLUX-Fill [3] takes approximately 9.7 seconds, whereas our refiner adds only 2.7 seconds, accounting for only 21.8% of the total inference time.

Notably, our approach remains more efficient than additional diffusion-based refinement stage. This efficiency stems from two factors: (1) PixPerfect requires only a single forward pass without iterative sampling, and (2) its architecture is lightweight and resolution-agnostic, enabling low-latency

execution. Even when inference-time pooling is enabled, the overall runtime remains within 1.3× of the baseline, while yielding measurable improvements in visual quality.

These results indicate that PixPerfect can be seamlessly integrated into existing diffusion pipelines with minimal computational cost, offering substantial perceptual gains at a fraction of the runtime.

## B.2 Comparisons with Poisson Blending

In the main paper, we have presented the comparisons with decoder-based method Assymetric VQ-GAN [65] and harmonization-based method DiffHarmony [64]. In this section we will provide additional analysis about Poisson blending. Poisson blending is a classical gradient-domain technique widely used for seamless image compositing. It estimates a smooth transition between a source (edited) region and a target (background) image by solving for pixel values that minimize gradient differences while respecting boundary conditions.

However, applying Poisson blending in the context of inpainting or local editing typically requires access to a reliable gradient field within the masked region. In practice, this is often approximated using the ground truth content in the masked area to compute the desired gradients. While this produces visually smooth results, it introduces a critical ground-truth leakage issue—information that is unavailable at test time is used during blending. Consequently, Poisson blending cannot be considered a fair or deployable baseline in real-world settings.

Although Poisson blending relies on inaccessible ground-truth information, we still present some qualitative comparison results. We apply Poisson blending on the outputs of FLUX-Fill [3] using ground-truth-masked gradients to simulate its ideal behavior. Fig. 7 shows representative examples comparing our method with Poisson blending. While the latter can reduce abrupt seams at the boundary, it often introduces unnatural hue propagation and fails to correct texture inconsistencies or geometric artifacts introduced during the generation process. Furthermore, when the inpainted results differ from the original ground truth image, the Poisson blending will blend the masked part into the tone of the original ground truth and produce unnatural seams. In contrast, our method produces more coherent integration with the background, better preserves structural details, and eliminates color/texture artifacts without relying on inaccessible ground-truth information.

These results highlight that Poisson blending falls short in correcting complex local editing artifacts. Our learning-based refiner not only avoids the pitfalls of ground-truth leakage but also achieves better perceptual quality through semantically aware refinement.

## B.3 More Qualitative results

To further illustrate the effectiveness and generalization of our approach, we present additional qualitative results for the two local editing tasks: object removal and object insertion. These tasks requires image editing within a masked area and keep the background unchanged.

In the object removal examples shown in Fig. 8, we present qualitative results from three representative baselines: OmniPaint [55], PowerPaint [66] and CLIPAway [11]. As indicated by the red arrows, baseline inpainting results often exhibit low-level inconsistencies, such as chromatic shifts, particularly in regions of clean background such as floors and tables. In contrast, our method effectively eliminates these artifacts, yielding smooth and contextually coherent background completions without disrupting the surrounding scene geometry.

In the object insertion results shown in Fig. 9, we visualize our refinement performance on outputs from ObjectStitch [41], AnyDoor [6], and PBE [50]. In these cases, challenges arise from the need to harmonize inserted objects with scene textures and lighting. As highlighted in the magnified insets, baseline results often suffer from blurry transitions, scale-inconsistent textures, or unnatural object boundaries. Our method noticeably improves local consistency by refining high-frequency texture alignment, enhancing boundary sharpness, and reducing chromatic discrepancies—leading to more realistic and visually pleasing composites.

Overall, these examples demonstrate the general applicability of our method across diverse models and editing scenarios. In both insertion and removal tasks, PixPerfect consistently enhances visual quality by resolving local inconsistencies that are challenging for latent diffusion models alone. We

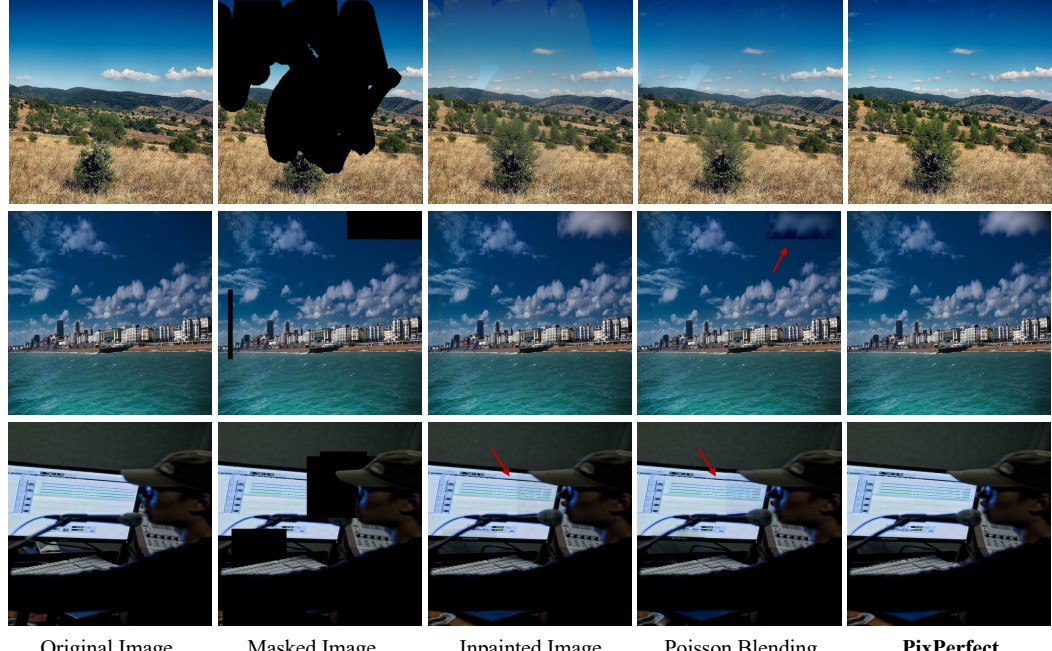

| Original Image | Masked Image | Inpainted Image | Poisson Blending | **PixPerfect** |

Figure 7: Qualitative comparison between our method and Poisson blending for FLUX-Fill inpainting outputs. While Poisson blending reduces edge discontinuities, it often introduces hue bleeding and fails to correct texture or structural artifacts. Further more in the cases where the inpainted results differ from the ground truth image (*e.g.* second row), poisson blending will tend to mimic the ground truth and produce unnatural results. In contrast, PixPerfect produces cleaner transitions, preserves scene structure, and avoids tone inconsistency without relying on inaccessible ground-truth information.

encourage readers to examine the highlighted regions closely to appreciate the subtle yet impactful improvements brought by our approach.

## C Implementation Details

**Architecture and Training.** The refiner is built on the CMGAN architecture [61]. However, we replace the bottleneck fully-connected layer with a global average pooling operation, thereby making the network fully convolutional. In addition, we apply channel pruning to reduce the model size. Our final model contains 41M parameters. Training employs R1 regularization with $\gamma = 1$ and utilizes the CoModGAN mask generation scheme [59] to generate random masks on-the-fly. During an initial warm-up phase, the discriminative pixel-space loss remains disabled. A constant learning rate of $5 \times 10^{-4}$ is applied throughout the training.

**Details on Color Shifting Augmentation.** Three complementary color-shifting schemes are employed. First, *linear gradient color augmentation* constructs a mask $\alpha$ by projecting normalized $x$–$y$ coordinate grids onto a randomly oriented unit vector and normalizing the result; the final image is obtained by alpha-blending this mask with a color-jittered version of the input. Second, *random blob color augmentation* synthesizes one or more soft ellipses per image—each defined by a random center, semi-axes sampled from a fraction of the image dimensions, and a random rotation—where pixel intensities decay smoothly from center to boundary; overlapping ellipses merge via a maximum operator to produce distinct, softly blended circular regions. Third, *uniform jitter augmentation* simulates spatially invariant color shifts by blending a uniformly color-jittered image with the original input using a fixed blending ratio. We provide an artifact generation pipeline that describes the artifact types and their corresponding augmentation probabilities in 5.

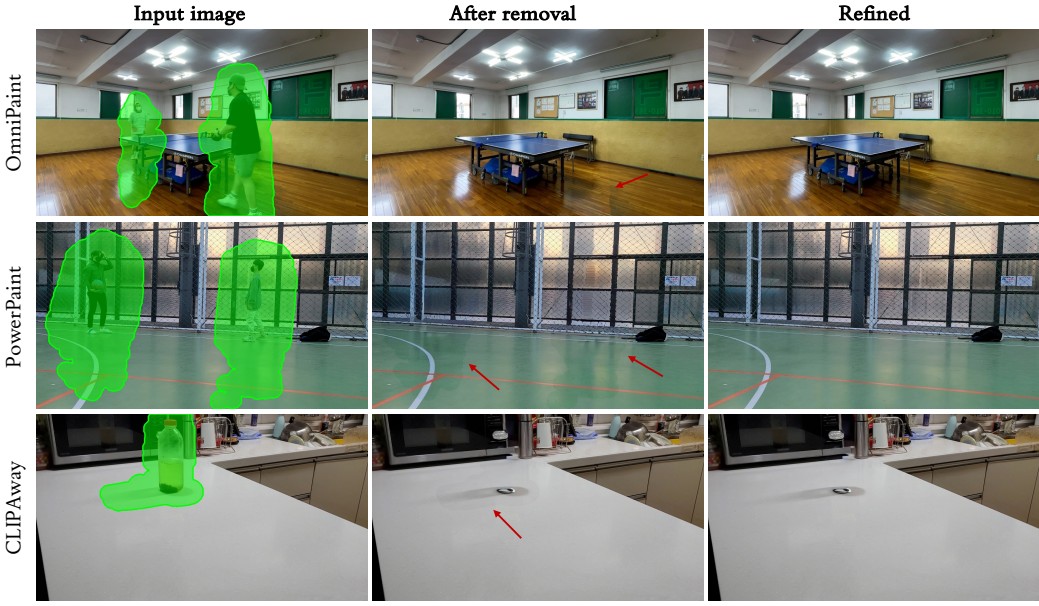

Figure 8: Qualitative comparisons on object removal. Red arrows highlight residual artifacts such as color inconsistency produced by baseline diffusion models. Our method effectively eliminates such artifacts.

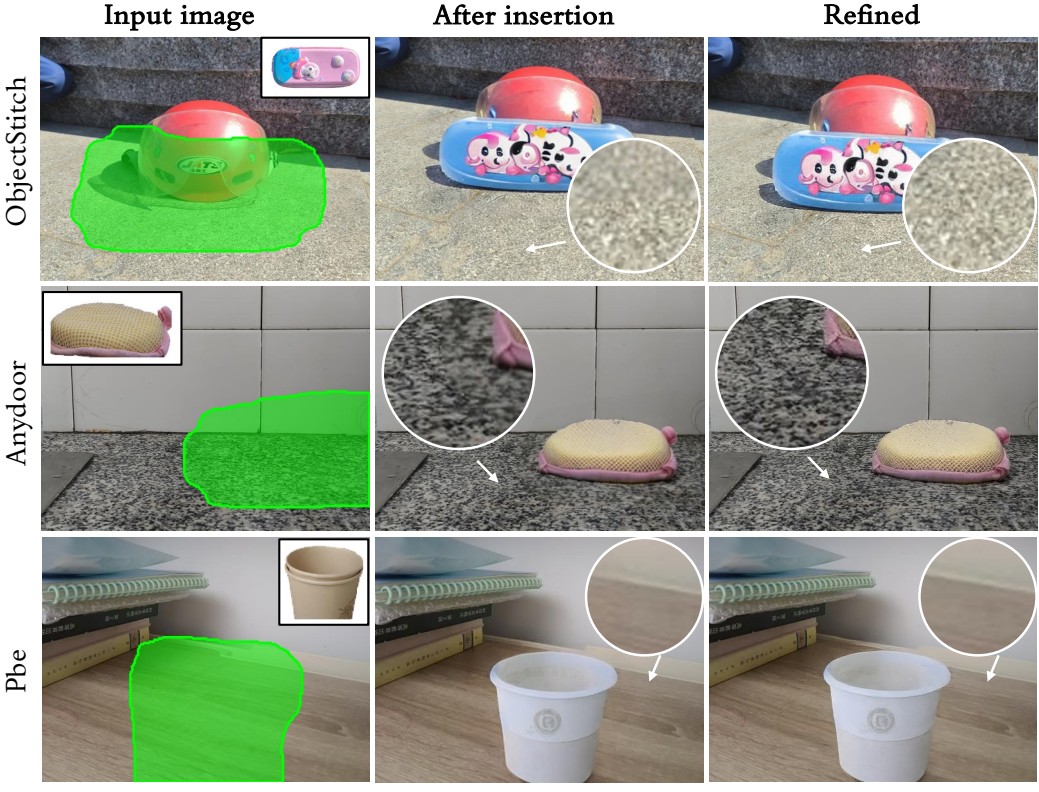

Figure 9: Qualitative comparisons on object insertion. The highlighted insets reveal artifacts in baseline results, such as blurry edges, inconsistent textures, and poor object blending. Our refinement enhances boundary sharpness, aligns local textures, and achieves more seamless visual integration.

Table 5: Summary of artifact types and their corresponding augmentation probabilities.

| Artifact Type | Description | Probability |
|---|---|---|
| Content Discontinuity | Small misalignments / missing pixels near mask edges | 0.5 |
| Background Color Augmentation | Non-uniform hue / brightness variations applied to the background | 0.8 |
| Foreground Color Augmentation | Non-uniform / uniform / gradient color perturbations applied to the foreground region | 0.8 |
| Soft/Hard Boundary Mixing | Mixing soft / hard boundaries to mimic visual seams at compositional borders | 1.0 |
| Sensor Noise / JPEG / Blur | Injecting noise / JPEG compression / blur into foreground and/or background regions | 0.5 |
| VAE Compression Artifacts | Introducing compression artifacts simulated by a pretrained VAE to the foreground | 0.5 |

**A minimal demo script for reproducing the "seam" artifacts of Flux inpainting [3] model.** To facilitate reproducibility, we attached a minimal demo script that reproduces the boundary artifacts for the official FLUX-Fill model [3].

```
1  import torch
2  import numpy as np
3  from PIL import Image, ImageDraw
4  from diffusers import FluxFillPipeline
5  from diffusers.utils import load_image
6
7  # === Define input image path ===
8  input_image_path = "/your/image/path" # TODO: change to the input image path
9
10 # === Load input image ===
11 image = load_image(input_image_path).convert("RGB")
12 width, height = image.size
13
14 # === Generate irregular mask ===
15 def generate_irregular_mask(width, height, max_shapes=5):
16     mask = Image.new("L", (width, height), 0)
17     draw = ImageDraw.Draw(mask)
18
19     for _ in range(np.random.randint(1, max_shapes + 1)):
20         shape_type = np.random.choice(["ellipse", "polygon"])
21         if shape_type == "ellipse":
22             x0, y0 = np.random.randint(0, width - 50), np.random.randint(0, height
    ↪ - 50)
23             x1, y1 = x0 + np.random.randint(40, 120), y0 + np.random.randint(40,
    ↪ 120)
24             draw.ellipse([x0, y0, x1, y1], fill=255)
25         else:
26             num_points = np.random.randint(3, 8)
27             points = [(np.random.randint(0, width), np.random.randint(0, height))
    ↪ for _ in range(num_points)]
28             draw.polygon(points, fill=255)
29
30     return mask.convert("RGB")
31
32 mask = generate_irregular_mask(width, height)
33
```

```python
34  # === Load FLUX inpainting pipeline ===
35  pipe = FluxFillPipeline.from_pretrained(
36      "black-forest-labs/FLUX.1-Fill-dev",
37      torch_dtype=torch.bfloat16
38  ).to("cuda")
39
40  # === Run FLUX-Fill ===
41  output = pipe(
42      image=image,
43      mask_image=mask,
44      prompt="",
45      height=height,
46      width=width,
47      guidance_scale=30, # The default value provided on the official huggingface page
48      num_inference_steps=50, # The default value provided on the official
        ↪ huggingface page
49      max_sequence_length=512 # The default value provided on the official
        ↪ huggingface page
50  ).images[0]
51
52  # === Composite: restore unmasked regions from original image ===
53  image_np = np.array(image)
54  output_np = np.array(output)
55  mask_np = np.array(mask.convert("L"))
56  inpainted_np = output_np.copy()
57  inpainted_np[mask_np < 128] = image_np[mask_np < 128]
58  inpainted = Image.fromarray(inpainted_np)
59
60  # === Save outputs ===
61  image.save("original.png")
62  mask.save("mask.png")
63  inpainted.save("inpainted.png")
```

**Code 1:** *A minimal demo script for reproducing the "seam" artifacts of Flux inpainting [3] model.*

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
