# OpenReview forum: "PixPerfect: Seamless Latent Diffusion Local Editing with Discriminative Pixel-Space Refinement"
_NeurIPS.cc/2025/Conference — NeurIPS 2025 poster_

### Official Review · Reviewer_W1zs · 2025-06-05

**Clarity:** 3
**Significance:** 4
**Originality:** 3
**Rating:** 5
**Confidence:** 3

**Summary:**

This paper proposes a refinement method for inpainted or local-edited images produced by various generative models. The approach involves training a GAN network on large-scale data with various types of simulated artifacts. Meanwhile, two types of loss functions, one in the pixel space and the other in a discriminative pixel space, are designed to optimize the network. The proposed method is compatible with various image generation models, demonstrating strong generalizability. It also offers certain advantages in terms of efficiency.

**Questions:**

* I have a question regarding the amount of data required for training: to what extent does the model's performance depend on the amount of data? After all, 300 million images is not a small dataset.

**Ethical Concerns:**

["NO or VERY MINOR ethics concerns only"]

**Final Justification:**

My concerns have been largely addressed. During the rebuttal stage, the authors provided thorough explanations. Reviewer wXb9 also raised questions regarding the dataset construction, and the authors gave detailed clarifications about the data composition. Overall, this paper achieves strong results with a simple approach, which is the key factor behind my high evaluation of it.

**Limitations:**

Yes.

**Paper Formatting Concerns:**

I have a minor suggestion regarding the formatting: for Table 1, it would be more intuitive to place the same models on adjacent rows, for example, SDv1.5 and SDv1.5-PixPerfect, since the paper is not comparing differences between different LDMs, but rather the effect of applying PixPerfect to the same LDM.

**Quality:**

3

**Strengths And Weaknesses:**

## Strengths
* The writing in this paper is smooth and clear, with rich detail throughout.
* The proposed approach demonstrates strong generalizability and can be applied to various image local-editing methods. Moreover, it exhibits excellent inference performance, which is highly practical for real-world applications.
* The experiments in this paper are relatively comprehensive, and the effectiveness of the proposed approach is significant from the perspective of both visualization and quantitative analysis.

## Weaknesses
* I wonder whether pixel-level inconsistencies along editing boundaries are actually a serious issue. If there are any relevant experiments or data, could you please provide them? Additionally, would simply achieving good performance with the LDM used for editing be sufficient to address this problem?

---

> ### Author Rebuttal · Authors · 2025-07-31
>
> We appreciate your positive and constructive feedback. We are especially grateful for the recognition of our work, including **strong generality**, **highly practical for real-world applications**, **comprehensive experiments**. We appreciate the opportunity to address the remaining questions.
>
> > 1. "I wonder whether pixel-level inconsistencies along editing boundaries are actually a serious issue. If there are any relevant experiments or data, could you please provide them? Additionally, would simply achieving good performance with the LDM used for editing be sufficient to address this problem?"
>
> We appreciate the reviewer for raising this important point. In our experiments, we found that the pixel-level inconsistency is a **systematic issue** that presists across a wide range of inpainting and editing models, including the most recent state-of-the-art methods such as FLUX-Fill [1] and OmniPaint [2]. In fact, the artifacts shown in **Figure 4** and **Figure 5** persist across most samples in the testing sets, showing clearly visible hue seams and texture discontinuities. To facilitate reproducibility, we have included demo code at the end of this rebuttal to allow reviewers to observe this pixel-level inconsistency.
>
> While improving the performance of the latent diffusion model (LDM) used for editing could potentially reduce the severity of these artifacts, we would like to emphasize that the **compression nature** of the latent space in LDMs inherently introduces distortion in low-level patterns. As a result, texture mismatches and noise artifacts remain difficult to avoid. Recent studies [3, 4] further demonstrate that the latent space is inherently spatially entangled, making it ill-suited for enforcing pixel-accurate boundary consistency. These observations underscore the need for a pixel-space framework that can explicitly address and correct inconsistencies at the pixel level.
>
> *[1] Black Forest Labs. FLUX.1 Fill: open-weight rectified-flow transformer for inpainting & outpainting, 2025.
> [2] Yu, Yongsheng, et al. "Omnipaint: Mastering object-oriented editing via disentangled insertion-removal inpainting.", ICCV'25.
> [3] Hou, Xingzhong, et al. "Towards Seamless Borders: A Method for Mitigating Inconsistencies in Image Inpainting and Outpainting." arXiv preprint arXiv:2506.12530 (2025).
> [4] Wang, Yikai, et al. "Towards Enhanced Image Inpainting: Mitigating Unwanted Object Insertion and Preserving Color Consistency." CVPR'25.*
>
> > 2. I have a minor suggestion regarding the formatting: for Table 1, it would be more intuitive to place the same models on adjacent rows, for example, SDv1.5 and SDv1.5-PixPerfect, since the paper is not comparing differences between different LDMs, but rather the effect of applying PixPerfect to the same LDM.
>
> Thank you for the suggestion. We will make sure to adjust the formatting to better highlight the effect of our method.
>
> ---
> **Demo code for reproducing FLUX-Fill seams**
> The following snippet provides a minimal setup for observing hue seams and texture discontinuities introduced by FLUX-Fill. Please ensure the model weights and dependencies are properly set up.
> ```python
> import torch
> import numpy as np
> from PIL import Image, ImageDraw
> from diffusers import FluxFillPipeline
> from diffusers.utils import load_image
>
> # === Define input image path ===
> input_image_path = "/your/image/path" # TODO: change to the input image path
>
> # === Load input image ===
> image = load_image(input_image_path).convert("RGB")
> width, height = image.size
>
> # === Generate irregular mask ===
> def generate_irregular_mask(width, height, max_shapes=5):
>     mask = Image.new("L", (width, height), 0)
>     draw = ImageDraw.Draw(mask)
>
>     for _ in range(np.random.randint(1, max_shapes + 1)):
>         shape_type = np.random.choice(["ellipse", "polygon"])
>         if shape_type == "ellipse":
>             x0, y0 = np.random.randint(0, width - 50), np.random.randint(0, height - 50)
>             x1, y1 = x0 + np.random.randint(40, 120), y0 + np.random.randint(40, 120)
>             draw.ellipse([x0, y0, x1, y1], fill=255)
>         else:
>             num_points = np.random.randint(3, 8)
>             points = [(np.random.randint(0, width), np.random.randint(0, height)) for _ in range(num_points)]
>             draw.polygon(points, fill=255)
>
>     return mask.convert("RGB")
>
> mask = generate_irregular_mask(width, height)
>
> # === Load FLUX inpainting pipeline ===
> pipe = FluxFillPipeline.from_pretrained(
>     "black-forest-labs/FLUX.1-Fill-dev",
>     torch_dtype=torch.bfloat16
> ).to("cuda")
>
> # === Run inpainting ===
> output = pipe(
>     image=image,
>     mask_image=mask,
>     prompt="",
>     height=height,
>     width=width,
>     guidance_scale=30, # The default value provided on the official huggingface page
>     num_inference_steps=50, # The default value provided on the official huggingface page
>     max_sequence_length=512 # The default value provided on the official huggingface page
> ).images[0]
>
> # === Composite: restore unmasked regions from original image ===
> image_np = np.array(image)
> output_np = np.array(output)
> mask_np = np.array(mask.convert("L"))
> inpainted_np = output_np.copy()
> inpainted_np[mask_np < 128] = image_np[mask_np < 128]
> inpainted = Image.fromarray(inpainted_np)
>
> # === Save outputs ===
> image.save("original.png")
> mask.save("mask.png")
> inpainted.save("inpainted.png")
> ```

---

> > ### Comment · Reviewer_W1zs · 2025-08-08
> > **Basically agree but still have a question about dataset scale**
> >
> > Thanks to the authors for the detailed explanation. The demo code provided for reproduction also offers more convincing evidence to demonstrate the existence of the issue of "pixel-level inconsistencies along editing boundaries". I basically agree with the author’s point of view. The compression nature of latent space in LDMs inherently results in distortion of low-level patterns. It is indeed difficult to avoid when compression exists in LDM. However, I still have one question: how does the dataset scale affect the performance of the approach? After all, 300 million images is not a small dataset.

---

> > > ### Author Response · Authors · 2025-08-08
> > > **About dataset scale**
> > >
> > > Thank you for the insightful question. Our refiner is a lightweight GAN-based model with 41M parameters, designed specifically to correct low-level artifacts such as color shifts, textures or noise patterns mismatch along editing seams. Since this low-level refinement task primarily preserves the structure of the input image rather than synthesizing new semantic content, it does not require a training set containing hundreds of millions of images. In addition, our data augmentation pipeline is designed to produce highly randomized and diversified variations of artifacts, which further reduces the need for extremely large datasets. In our experiment, we train the model on our in-house dataset, which happened to contain approximately 300M images, but our approach does not depend on this scale. A medium-scale dataset such as Places365-Challenge [1], combined with our augmentation pipeline, is also viable for this task.
> > >
> > > [1] Zhou, B., Lapedriza, A., Khosla, A., Oliva, A., & Torralba, A. (2017). Places: A 10 million Image Database for Scene Recognition. *IEEE Transactions on Pattern Analysis and Machine Intelligence*. IEEE.

---

> ### Author Response · Authors · 2025-08-06
> **Discussion Reminder**
>
> Dear Reviewer W1zs,
>
> We noticed that you have not reply to the authors' rebuttal and we are currently not be able to see your response. We hope we have adequately addressed all your concerns in our previous discussions. We would be happy to continue the discussion here if there are any remaining points you would like us to clarify.
>
> Best regards, The Authors

---

### Official Review · Reviewer_JDA3 · 2025-07-01

**Clarity:** 4
**Significance:** 3
**Originality:** 3
**Rating:** 5
**Confidence:** 4

**Summary:**

This paper introduces a refining method for generative in- and outpainting approaches. Recent generative models shine on conditional image generation, but they often struggle with pixel-perfect in- and outpainting, often leading to noticeable artifacts inside the masked region. The authors propose to solve these artifacts with a pixel-space refiner module, trained with various image degradations. Additionally, a pixel space color discriminator is inroduced to help localizing subtle color mismatches between the edited and original regions.

The proposed method can be applied to any backbone inpainting methods and consistenly improves the inpatining quality.

**Questions:**

## Figures
* Figure 1: What does the dashed arrow to the decoder in latent space means?

**Ethical Concerns:**

["NO or VERY MINOR ethics concerns only"]

**Final Justification:**

The paper provides a simple way to improve upon a heavily used downstream task of diffusion models: inpainting; making it interesting for a wide community. My only remaning concern is to see an ablation on the disciminative space dimensionality, which should be possible to show in the final revision.

**Limitations:**

yes

**Paper Formatting Concerns:**

No concerns.

**Quality:**

3

**Strengths And Weaknesses:**

## Strengths
* **S1** The proposed method is simple and agnositc to the backbone.
* **S2** Impressive quality with visible and clear improvement over previous methods in a comprehensive comparison.
* **S3** Well-written paper, easy to follow and understand.
* **S4** The tackled task is interesting for a broad community.

## Weaknesses
* **W1 - Ablating design choices** Although the authors did a good job on evaluating the key components of the method, it would be interesting to see if how the quality changes with the specific design choices. E.g. what if the discriminative pixel space is not 3 dimensional, but more (or less)?

---

> ### Author Rebuttal · Authors · 2025-07-31
>
> We sincerely thank the reviewer for the valuable comments and we are grateful for your recognition of the merits of our methods, including **simple and backbone-agnostic design**, **impressive quality with clear improvement**. We also appreciate your acknowledgment that the task we address is **of interest to a broad community**.
>
> Based on the positive assessment, we would like to respectfully elaborate on the remaining questions.
>
> > 1. It would be interesting to see if how the quality changes with the specific design choices. E.g. what if the discriminative pixel space is not 3 dimensional, but more (or less)?
>
> Thank you for this insightful suggestion. Inspired by the classic color space transformation [1,2], we adopt a direct mapping from the standard RGB space to a discriminative pixel space of equal dimensionality. This keeps the transformation both simple and easily interpretable.
>
>
> Here we present the ablation results of design choices for (1) different polynomial regression degrees (2) other methods such as HAAR.
>
> | Method | FID $\downarrow$ | LPIPS $\downarrow$ | L1 $\downarrow$ |
> |-------|-------|-------|------|
> | 6-degree regression loss **（PixPerfect）** | 10.8675 | 0.1414 | 0.0363 |
> | 2-degree regression loss  | 11.2244 | 0.1431 | 0.0362 |
> | 10-degree regression loss | 11.0018 | 0.1407 | 0.0361 |
> | HAAR loss  | 11.3816 | 0.1431 | 0.0375 |
>
>
> It will be interesting to test other designs. This can be left as future work.
>
>
> > 2. What does the dashed arrow to the decoder in latent space means?
>
> The dashed arrow denotes an optional pathway where in certain design variants, the decoder can take the original image as an additional condition to improve background consistency [3]. We will clarify this in the final version.
>
> *[1] CIE. *Uniform Color Spaces (CIELUV)*. Supplement 2 to **CIE 15: Colorimetry**, 1978.*
> *[2] Smith, A. R. *Color Gamut Transform Pairs*. SIGGRAPH Proceedings, 1978 (introduces HSV).*
> *[3] Z. Zhu, et al. Designing a better asymmetric vqgan for stablediffusion, arXiv:2306.04632, 2023.*

---

> > ### Comment · Reviewer_JDA3 · 2025-08-04
> > **Still positive, expect one ablation in revision**
> >
> > I thank the authors for the provided ablation on the regression loss. Overall, I am still positive about the paper, since it provides a simple way to improve upon a heavily used downstream task of diffusion models: inpainting. I see that a 3-channel discriminative space is easier to interpret, but higher dimensions can still be projected to 3 for inspection purposes, like using PCA. Therefore, I hope to see an ablation on the pixel space dimensionality in the revision of the paper.

---

> > > ### Author Response · Authors · 2025-08-06
> > > **Detail clarification on the high-dimensional space**
> > >
> > > We would like to express our thanks for the positive assessment and for your suggestion regarding high-dimensional discriminative spaces. We will add an ablation on the high-dimensional pixel space and report the corresponding performance. To ensure we match your intent: are you suggesting applying the discriminative-space transformation to a d-dimensional embedding (e.g., a perceptual / VGG feature space) rather than to the RGB space itself? Thanks again!

---

> > > > ### Comment · Reviewer_JDA3 · 2025-08-06
> > > > **Clarification on the high-dimensional space**
> > > >
> > > > Yes, I mean to use a higher dimensional discriminative space, not RGB.

---

> > > > > ### Author Response · Authors · 2025-08-09
> > > > > **Ablation on high-dimensional discriminative space**
> > > > >
> > > > > Thank you for raising this insightful point. We implemented a variant of our loss that applies the discriminative transformation to VGG16 feature maps before computing the LPIPS loss. When trained with this high-dimensional discriminative space variant, our model achieves competitive performance (see table below), suggesting that our method is generalizable to a higher-dimensional discriminative space. However, its performance is slightly worse than the original 3-dimensional implementation, which we hypothesize is due to the loss of spatial precision along editing boundaries caused by spatial downsampling of feature maps. We will include this ablation in the revised version of the paper and are interested in further exploring and high-dimensional discriminative spaces in future work. We sincerely appreciate your valuable feedback!
> > > > >
> > > > > | Method | FID $\downarrow$ | LPIPS $\downarrow$ | L1 $\downarrow$ |
> > > > > |-------|-------|-------|------|
> > > > > | the 3-dimensional loss **（PixPerfect）** | 10.8675 | 0.1414 | 0.0363 |
> > > > > | the high-dimensional loss | 11.0525 | 0.1421 | 0.0360 |

---

### Official Review · Reviewer_x4mB · 2025-07-01

**Clarity:** 4
**Significance:** 3
**Originality:** 3
**Rating:** 4
**Confidence:** 3

**Summary:**

This paper introduces PixPerfect, a pixel-level refinement framework designed to address artifacts from local edits by inpainting models. The pipeline includes standard latent space inpainting, a pixel-space refiner, and a discriminative pixel space to enhance and correct subtle color and texture inconsistencies. Additionally, the authors propose an artifact simulation pipeline during training, covering non-uniform color shifts, texture mismatches, and content discontinuities. Extensive experiments and ablation studies are presented to demonstrate the effectiveness of the method.

**Questions:**

I see the responses to my two main concerns: (1) why do we need to use regression, rather than FFT or HAAR? (2) the baseline quality of FLUX-Fill.

Besides, I have some further questions or small points:
* Is the trained refiner a general module that works across different LDM inpainting models, or are there separate refiners for each model?
* Since the training uses the synthetic artifact simulation pipeline, does this mean the LDM inpainting model is not involved during the refiner's training?
* The font size in the tables is inconsistent, for example, in Table 4.
* What base model is used in Figure 5?

**Ethical Concerns:**

["NO or VERY MINOR ethics concerns only"]

**Final Justification:**

The rebuttal by the authors has addressed my concerns. I decide to increase my rating as a positive score.

**Limitations:**

yes

**Quality:**

3

**Strengths And Weaknesses:**

Strengths
* The paper is well-written and easy to understand.
* The motivation is clear, focusing on solving artifacts caused by LDM models during the inpainting process.
* The synthetic artifact simulation pipeline is well-designed and appears effective.
* The evaluations are comprehensive, covering benchmarks like MISATO, Places2, and RORDS, and comparing against SOTA methods.

Weaknesses
* I understand that the purpose of $y_{amp}$ is to highlight error maps and focus more on problematic areas. However, it is unclear why the authors chose polynomial regression with amplified attention instead of using frequency-domain techniques like FFT or HAAR, which could also re-weight error areas. This choice is not explained or validated through ablation studies.
* I also have doubts about the results in Figure 4. From my experience with FLUX-Fill, the boundary artifacts shown in column 2 don't seem as severe and obvious. I tested FLUX-Fill models during my review and did not observe such artifacts. Could you clarify the hyper-parameter settings, such as guidance scale or controlling strength, used during inference?
* If FLUX-Fill already produces decent results, the practical impact of the proposed method may be limited.

---

> ### Author Rebuttal · Authors · 2025-07-31
>
> We sincerely thank the reviewer for the valuable feedback and appreciate your recognition of our contributions, including **clear motivation**, **a well-designed simulation pipeline** and **comprehensive benchmark**. We will address all raised concerns as follows.
>
> > 1. I understand that the purpose of  is to highlight error maps and focus more on problematic areas. However, it is unclear why the authors chose polynomial regression with amplified attention instead of using frequency-domain techniques like FFT or HAAR, which could also re-weight error areas. This choice is not explained or validated through ablation studies.
>
> Thank you for raising this insightful point. In early experiments we did test frequency-domain re-weighting: we decomposed prediction image with HAAR decomposition, amplified the low-frequency sub-bands, and used the recombined predicted image to compute the training loss. However, this approach consistently produced overly smooth, blurry results. In fact, the unequal scaling of sub-bands disrupted the balance between low- and high-frequency components, leading disrupted textures and noise patterns, thus hinding the network to learn to predict the correct textures that matchs the ground truth.
>
> We further examined frequency-domain decomposition by training a model that employs a Haar-based re-weighted loss. In this variant, the prediction and ground truth are each decomposed into low- and high-frequency bands; separate $\ell_{1}$ losses are computed for the two bands, and the low-frequency term is assigned a larger weight to target subtle colour shifts. As the following table shows, this approach performs worse than our method. Moreover, band-wise re-weighting is feasible with losses such as $\ell_{1}$ or $\ell_{2}$; it is incompatible with perceptual or GAN objectives, which are shown crucial for image synthesis.
>
> | Method | FID $\downarrow$ | LPIPS $\downarrow$ | L1 $\downarrow$ |
> |-------|-------|-------|------|
> | ours loss **（PixPerfect）** | 10.8675 | 0.1414 | 0.0363 |
> | Haar-based re-weighted loss | 11.3816 | 0.1431 | 0.0375 |
>
> By contrast, the polynomial-regression we ultimately adopted is an **element-wise transformation** in pixel space. Because it modifies each pixel without altering the spatial relationship with neighbouring pixels, it preserves subtle texture and noise patterns while still amplfying subtle color difference.
>
>
> > 2. I also have doubts about the results in Figure 4. From my experience with FLUX-Fill, the boundary artifacts shown in column 2 don't seem as severe and obvious. I tested FLUX-Fill models during my review and did not observe such artifacts. If FLUX-Fill already produces decent results, the practical impact of the proposed method may be limited.
>
> We sincerely thank the reviewer for taking the time to run hands-on tests. In addition, we would like to clearify our evaluation setting. In our experiment, we use the official FLUX-Fill model and the default hyper-parameters running on 512px resolution and paste the original background (Line 32, 254), following the standardized inpainting setting. We found that FLUX-Fill often produce inconsistent color or textures patterns that are different from the input background. In fact, the artifacts shown in **Figure 4** persist across most samples in the testing sets. *To facilitate reproducibility, we attached a minimal demo script at the end of this rebuttal that reproduces the boundary artifacts for the official FLUX-Fill model.*
>
> We want to further emphasize that such issues are **common** and **fundemantal** challenges for LDM local editing models. The compression nature of the latent space in LDMs inherently introduces distortion in low-level patterns. As a result, texture mismatches and noise artifacts remain difficult to avoid. Recent studies [1,2] further demonstrate that the latent space is inherently spatially entangled, making it ill-suited for enforcing pixel-accurate boundary consistency. These observations underscore the need for a pixel-space framework that can explicitly address and correct inconsistencies at the pixel level.
>
> *[1] Hou, Xingzhong, et al. "Towards Seamless Borders: A Method for Mitigating Inconsistencies in Image Inpainting and Outpainting." arXiv preprint arXiv:2506.12530 (2025).*
> *[2] Wang, Yikai, et al. "Towards Enhanced Image Inpainting: Mitigating Unwanted Object Insertion and Preserving Color Consistency." CVPR'25.*
>
> > 3. Is the trained refiner a general module that works across different LDM inpainting models, or are there separate refiners for each model?
>
> Thank you for the question. The refiner is a single, model-agnostic module. It is trained once and then applied without any additional fine-tuning to all LDM-based inpainting and local-editing backbones evaluated in the paper.
>
> > 4. Since the training uses the synthetic artifact simulation pipeline, does this mean the LDM inpainting model is not involved during the refiner's training?
>
> Yes. Our artifact simulation pipeline is agnostic to different diffusion-model. This design enables our trained refiner to serve as a general-purpose refinement module for various LDM-based editing pipelines without retraining. (L166-L169)
>
> > 5. What base model is used in Figure 5?
>
> As indicated in the left caption of Figure 5, the base model is OmniPaint for object insertion and Anydoor for object removal.
>
>
>
> > 6. The font size in the tables is inconsistent, for example, in Table 4.
>
> Thanks for pointing out. We will fix this issue in the final version.
>
> ---
> *Demo code for reproducing FLUX-Fill seams*
> The following snippet provides a minimal setup for observing hue seams and texture discontinuities introduced by FLUX-Fill. Please ensure the model weights and dependencies are properly set up.
>
> ```python
> import torch
> import numpy as np
> from PIL import Image, ImageDraw
> from diffusers import FluxFillPipeline
> from diffusers.utils import load_image
>
> # === Define input image path ===
> input_image_path = "/your/image/path" # TODO: change to the input image path
>
> # === Load input image ===
> image = load_image(input_image_path).convert("RGB")
> width, height = image.size
>
> # === Generate irregular mask ===
> def generate_irregular_mask(width, height, max_shapes=5):
>     mask = Image.new("L", (width, height), 0)
>     draw = ImageDraw.Draw(mask)
>
>     for _ in range(np.random.randint(1, max_shapes + 1)):
>         shape_type = np.random.choice(["ellipse", "polygon"])
>         if shape_type == "ellipse":
>             x0, y0 = np.random.randint(0, width - 50), np.random.randint(0, height - 50)
>             x1, y1 = x0 + np.random.randint(40, 120), y0 + np.random.randint(40, 120)
>             draw.ellipse([x0, y0, x1, y1], fill=255)
>         else:
>             num_points = np.random.randint(3, 8)
>             points = [(np.random.randint(0, width), np.random.randint(0, height)) for _ in range(num_points)]
>             draw.polygon(points, fill=255)
>
>     return mask.convert("RGB")
>
> mask = generate_irregular_mask(width, height)
>
> # === Load FLUX inpainting pipeline ===
> pipe = FluxFillPipeline.from_pretrained(
>     "black-forest-labs/FLUX.1-Fill-dev",
>     torch_dtype=torch.bfloat16
> ).to("cuda")
>
> # === Run FLUX-Fill ===
> output = pipe(
>     image=image,
>     mask_image=mask,
>     prompt="",
>     height=height,
>     width=width,
>     guidance_scale=30, # The default value provided on the official huggingface page
>     num_inference_steps=50, # The default value provided on the official huggingface page
>     max_sequence_length=512 # The default value provided on the official huggingface page
> ).images[0]
>
> # === Composite: restore unmasked regions from original image ===
> image_np = np.array(image)
> output_np = np.array(output)
> mask_np = np.array(mask.convert("L"))
> inpainted_np = output_np.copy()
> inpainted_np[mask_np < 128] = image_np[mask_np < 128]
> inpainted = Image.fromarray(inpainted_np)
>
> # === Save outputs ===
> image.save("original.png")
> mask.save("mask.png")
> inpainted.save("inpainted.png")
> ```

---

> ### Author Response · Authors · 2025-08-06
> **Discussion Reminder**
>
> Dear Reviewer x4mB,
>
> We noticed that you have not reply to the authors' rebuttal and we are currently not be able to see your response. We hope we have adequately addressed all your concerns in our previous discussions. We would be happy to continue the discussion here if there are any remaining points you would like us to clarify.
>
> Best regards, The Authors

---

> ### Comment · Reviewer_x4mB · 2025-08-06
> **Response to authors**
>
> Thank you for providing detailed explanations and testing code. I have tried a few simple cases and noticed some boundary issues. While techniques like blend diffusion can help mitigate these problems, I agree that employing a specialized model to address such issues could be another feasible and meaningful research direction. At this point, I am inclined to increase my rating, provided no additional concerns are raised by other reviewers.

---

> > ### Author Response · Authors · 2025-08-09
> >
> > Thank you for carefully testing our code and for recognizing our approach as a feasible and meaningful research direction. We agree that diffusion-side techniques (e.g., blend diffusion, LayerDiffuse) can help mitigate boundary issues. However, the **compression** inherent to the latent space in LDMs often introduces distortion in low-level patterns, leading to texture and noise patterns mismatches that are difficult to avoid with diffusion techniques alone. A possible future direction is to combine a pixel-space model with diffusion-side techniques to further enhance boundary quality.
> >
> > *We have also carefully addressed the concerns and questions raised by all other reviewers.* We greatly appreciate your positive assessment and openness to increasing the rating.

---

### Official Review · Reviewer_wXb9 · 2025-07-05

**Clarity:** 3
**Significance:** 3
**Originality:** 3
**Rating:** 3
**Confidence:** 4

**Summary:**

This paper introduces PixPerfect, a pixel-level refinement framework designed to address pixel-level inconsistencies (such as chromatic shifts, texture mismatches, and visible seams) that occur in local editing tasks using latent diffusion models (LDMs). The paper constructs a comprehensive artifact simulation pipeline to generate a large amount of diverse data and maps images from the RGB pixel space to a differentiable discriminative pixel space, amplifying the differences between the mask region and the original background region. This paper improves the performance of LDMs models in local editing tasks from both the data and algorithm levels.

**Questions:**

1. In Section 3.1, when designing the tone mapping function, the authors directly consider the polynomial regression approach. Is there any theoretical validation for this approach? Could you provide some references?

2. In lines 151-152, the authors mention that this tone mapping function is a closed-form, sample-specific, differentiable mapping. What is the data volume based on which this tone mapping function is determined? Do the parameters of this mapping function need to be specially adjusted for each dataset? How to ensure the generalization ability of this mapping function?

3. Can the parameters of this tone mapping function be adjusted? Will the degree of the polynomial affect the final result? There seems to be no relevant ablation experiment later.

4. In Section 4.1, the authors state that they used a data volume of 300M. Are all these data generated through the artifact data production pipeline? Could you explain the data volume ratio of each artifact type?

**Ethical Concerns:**

["NO or VERY MINOR ethics concerns only"]

**Limitations:**

See questions

**Quality:**

3

**Strengths And Weaknesses:**

Advantage：
1.The scenario studied in this paper is highly novel. Although it is relatively specific, it is crucial to the final effect of image editing. The proposed PixPerfect model successfully solves the problem of artifacts generated in image editing based on latent diffusion models (LDMs) and effectively improves the effect of image editing.
2.In terms of algorithms, this paper proposes that using a discriminative pixel space during training can effectively solve the artifact problem in image editing. This is a simple and direct method, which only needs to add the GAN + reconstruction loss in the discriminative pixel space to the GAN + reconstruction loss obtained in the standard pixel space, and the dimensions of the standard pixel space and the discriminative pixel space are the same.
3.The synthetic data pipeline proposed in this paper focuses on four aspects: non-uniform color shifting, texture-pattern mismatch, content discontinuities, and mixing soft and hard boundaries, which basically covers the common artifact problems in image editing.

---

> ### Author Rebuttal · Authors · 2025-07-31
>
> We sincerely thank the reviewer for the detailed feedback. We are encouraged that the reviewer found that the reviewer found the **scenario studied is highly novel and crucial for image editing** and our method **better than or competitive with existing methods.** We appreciate the opportunity to address the concerns raised and clarify the details.
>
> ### Regarding on Weaknesses and Questions
>
> > 1. In Section 3.1, when designing the tone mapping function, the authors directly consider the polynomial regression approach. Is there any theoretical validation for this approach? Could you provide some references?
>
> The tone mapping function is designed to construct a discriminative pixel space that amplifies subtle inconsistencies. In the literature, classical color space transformations such as CIELUV and HSV [1,2] are established examples of non-linear remappings that make chromatic differences more perceptually uniform. Following this principle, we introduce the sample-adaptive low-degree polynomial mapping, which naturally emphasizes discrepancies at editing seams while minimally affecting consistent regions. This polynomial formulation ensures a closed-form solution and is fully differentiable, making it suitable for integration into the training pipeline.
>
>
> *[1] CIE. Uniform Color Spaces (CIELUV). Supplement 2 to CIE 15: Colorimetry, 1978.*
> *[2] Smith, A. R. Color Gamut Transform Pairs. SIGGRAPH Proceedings, 1978 (introduces HSV).*
>
> > 2. In lines 151–152, the authors mention that this tone mapping function is a closed-form, sample-specific, differentiable mapping. What is the data volume based on which this tone mapping function is determined? Do the parameters of this mapping function need to be specially adjusted for each dataset? How to ensure the generalization ability of this mapping function?
> >
>
> To clarify, the tone mapping function is not trained over a dataset, but is dynamically computed per image during training, based on a closed-form polynomial regression fitted to a set of pixels sampled from the foreground (masked) and background (unmasked) regions of that specific image. This sample-specific design ensures that the mapping always highlights the actual artifact region for in each training instance in a dynamic fashion, making the learning signal highly relevant and robust.
>
> Since the mapping is calculated with a closed-form solution for each individual sample, it does not require manual tuning or parameter adjustment across datasets. Its role is to amplify chromatic and textural differences in a differentiable way, providing a stronger training gradient for the refinement network. Generalization is achieved through the refiner, which learns to correct a wide range of artifact patterns, while the tone mapping serves as an auxiliary projection to emphasize discrepancies more effectively than conventional RGB losses.
>
> > 3. Can the parameters of this tone mapping function be adjusted? Will the degree of the polynomial affect the final result? There seems to be no relevant ablation experiment later.
>
> Yes, the parameters of the polynomial tone mapping can be adjusted. Our empirical experiments show that low polynomial degree results only mild tonal adjustment, rendering the loss less effective. Conversely, higher degree tends to over-amplify details and produces images that deviate from the natural image distribution, thus hurts the performance. We therefore select a moderate degree as a practical trade-off. We also present the ablation results here.
>
> | Method | FID $\downarrow$ | LPIPS $\downarrow$ | L1 $\downarrow$ |
> |-------|-------|-------|------|
> | 6 degrees **（PixPerfect）** | 10.8675 | 0.1414 | 0.0363 |
> | 2 degrees | 11.2244 | 0.1431 | 0.0362 |
> | 10 degrees | 11.0018 | 0.1407 | 0.0361 |
>
> > 4. In Section 4.1, the authors state that they used a data volume of 300M. Are all these data generated through the artifact data production pipeline? Could you explain the data volume ratio of each artifact type?
>
> Thanks for raising this point. Yes, our artifact simulation pipeline is a function that can run efficiently during training, thus all the data are generated through the pipeline.
> The artifact simulation pipeline is implemented by stochastically applying one or more artifact type in a sequential order. The type and probability of the operations in the following table. We will provide a detailed pseudo code in revision.
>
> | Artifact Type               | Description                                                        | Probability |
> |-----------------------------|--------------------------------------------------------------------|-------------|
> | Content Discontinuity       | Small misalignments or missing pixels near mask edges              | 0.5       |
> | Background Color Augmentation  | Non-uniform hue / brightness            | 0.8      |
> | Foreground Color Augmentation  | Non-uniform / uniform / gradient color augmentation           | 0.8      |
> | Soft/Hard Boundary Mixing      | Mixing soft and hard boundary to mimic seams           | 1.0      |
> | Adding sensor noise / JPEG artifacts / blurring | Adding noise to foreground and/or background         | 0.5     |
> | Adding VAE compression artifacts | Adding VAE compression artifact to foreground         | 0.5     |

---

> ### Author Response · Authors · 2025-08-06
> **Discussion Reminder**
>
> Dear Reviewer wxB9,
>
> We noticed that you have not reply to the authors' rebuttal and we are currently not be able to see your response. We hope we have adequately addressed all your concerns in our previous discussions. We would be happy to continue the discussion here if there are any remaining points you would like us to clarify.
>
> Best regards, The Authors

---

> > ### Author Response · Authors · 2025-08-08
> >
> > Dear Reviewer wXb9,
> >
> > Thank you once again for taking the time to review our paper. As the discussion deadline approaches, we would like to kindly draw your attention to our rebuttal and hear if there are any remaining questions or concerns. We will do our best to respond.
> >
> > If you have any feedback on the paper or our response, we would greatly appreciate it if you could leave a comment.  If you feel your concerns have been addressed, we would greatly value your updated reviews on the paper.
> >
> > Thank you very much for your attention.
> >
> > Best regards,
> > The Authors

---

> ### Comment · Area_Chair_XkZ7 · 2025-08-08
> **NeurIPS Author–Reviewer Discussion Deadline Tomorrow**
>
> Dear Reviewers,
>
> This is a gentle reminder that the author–reviewer discussion period ends tomorrow.
> If you have not yet engaged in discussion with the authors, please do so promptly. Constructive exchanges at this stage are critical to ensuring that all relevant clarifications and rebuttals are considered.
>
> Please also finalize your ratings and comments after the discussion period concludes, reflecting any changes in your assessment. Your timely participation will help ensure a fair and thorough review process.
>
> Thank you for your efforts and dedication.

---

### Note · Authors · 2025-08-15

We thank the AC and reviewers for the constructive feedback and engagement. Our submission received positive initial ratings, with broad recognition for its strong motivation, novel design, impressive performance, and thorough evaluation.
- wXb9 valued our *novel scenario*, *effectiveness in solving LDM artifact issues*, *simple algorithmic design*, and *comprehensive artifact simulation pipeline*.
- x4mB highlighted the *clear motivation*, *well-designed synthetic artifact pipeline*, and *comprehensive benchmarks*.
- JDA3 praised the *simple, backbone-agnostic design*, *visible improvements over prior methods*, and the *relevance to a broad community*.
- W1zs commended *the strong generalizability*, *practical applicability*, and *significant qualitative and quantitative improvements*.

During rebuttal, we thoroughly addressed the remaining technical and experimental concerns. Specifically, we:
- **Added more ablations** by including results on polynomial degree, a HAAR/FFT-style frequency re-weighting baseline, and a high-dimensional discriminative-space variant.
- **Released reproducibility resources** with a minimal script to reproduce FLUX-Fill boundary artifacts under the official settings, along with full clarification of all inference hyper-parameters.
- **Clarifying methodological design** by detailing the per-image closed-form tone-mapping rationale, polynomial degree selection, and parameterization.
- **Clarified data-scale dependence** by explaining the 41M-parameter refiner design and why our task does not rely on large scale image corpus.
- **Addressed presentation details**, noting formatting and figure adjustments to be incorporated in the camera-ready version.

Post-discussion outcome
Two reviewers (**JDA3**, **W1zs**) **consistently maintained positive ratings** (5-accept) after all questions were addressed.
Of the two initial boarderline (3-BR) reviewers, **x4mB shifted to a positive stance** after we provided reproducible experiments demonstrating that FLUX-Fill indeed exhibits the boundary artifacts we target, along with additional ablation over HAAR alternatives. The reviewer explicitly indicated willingness to raise the score. The other reviewer (**wXb9**) **raised only technical clarification questions in initial assessment and did not participate in the discussion phase**. These clarifications questions were fully addressed in the rebuttal, leaving no unresolved concerns.

---

### Decision · Program_Chairs · 2025-09-17

**Decision:**

Accept (poster)

**Comment:**

This paper introduces PixPerfect, a pixel-level refinement framework for addressing artifacts in local image editing tasks with latent diffusion models. After reviewing the rebuttal, all participating reviewers agree that the authors have adequately addressed concerns regarding the theoretical motivation for the polynomial tone-mapping function, the sample-specific and differentiable nature of the mapping, and the training data composition. Remaining minor points, such as ablations on the dimensionality of the discriminative pixel space and clarification of baseline hyperparameters, are suggested for inclusion in the final version. Although reviewer wXb9 did not participate in the discussion, the initial review also highlighted the novelty of the paper.

Overall, the paper presents a novel and practically relevant solution, with strong empirical results and a well-designed training pipeline. The AC therefore recommends acceptance, encouraging the authors to incorporate additional ablation studies and clarifications in the final version.